# Rapid diagnosis of skin and soft tissue melioidosis in children

Keang Suy[1,2*], Seda Bott[1], Nara Leng[3], Vuthy Sar[3], Sona Soeng[1], Sophanith Real[1], David AB Dance[4,5], Sue J. Lee[6,7,8], Paul Turner[1,6], Clare L. Ling[1,6☋], Arjun Chandna[1,6,9☋]

1 Cambodia Oxford Medical Research Unit, Angkor Hospital for Children, Siem Reap, Cambodia, 2 Medical Department, Angkor Hospital for Children, Siem Reap, Cambodia, 3 Surgical Unit, Angkor Hospital for Children, Siem Reap, Cambodia, 4 Lao Oxford Mahosot Hospital Wellcome Trust Research Unit, Mahosot Hospital, Vientiane, Lao PDR, 5 Faculty of Infectious and Tropical Diseases, London School of Hygiene and Tropical Medicine, London, United Kingdom, 6 Centre for Tropical Medicine & Global Health, University of Oxford, Oxford, United Kingdom, 7 Mahidol-Oxford Tropical Medicine Research Unit, Mahidol University, Bangkok, Thailand, 8 Department of Infectious Diseases, Alfred Hospital and School of Translational Medicine, Monash University, Melbourne, Australia, 9 Clinical Research Department, London School of Hygiene & Tropical Medicine, London, United Kingdom

☋ These authors contributed equally to this work.
* suykeang@angkorhospital.org

## Abstract

Melioidosis is an endemic infection caused by *Burkholderia pseudomallei*, found in tropical and subtropical regions. In resource-limited settings, culture-based diagnostics are often slow, delaying appropriate treatment, or unavailable. We conducted an interventional ambispective cohort study to assess the impact and diagnostic accuracy of the Active Melioidosis Detect Plus rapid diagnostic test (AMD-RDT) in Cambodian children with suspected skin and soft tissue melioidosis (SST-M). The retrospective cohort (July 2022–July 2023) received standard diagnostics; the prospective cohort (July 2023–December 2024) included AMD-RDT testing. Twenty-five retrospective culture-confirmed participants and 107 participants (31 culture-confirmed) in the prospective arm were analysed. Median time from pus collection to appropriate antibiotic initiation was 118.4 hours in the retrospective arm and 14.4 hours in the prospective arm (p = 0.057). Disseminated melioidosis workups were completed for 24% (6/25) and 80.6% (25/31) of retrospective and prospective participants respectively (p < 0.001), and detected two children with bacteraemia and three with intra-abdominal abscesses in the prospective arm. The AMD-RDT achieved a sensitivity of 90.3% (95% CI: 74.2–98.0%), specificity of 100% (95% CI: 95.3–100%), and an area under the receiver operating characteristic curve (AUC) of 0.95 (95% CI 0.89-1.00). Incorporating the AMD-RDT into the routine diagnostic pathway was associated with a reduction in time to effective antibiotic therapy and an increase in the proportion of participants completing a comprehensive diagnostic workup for systemic involvement. The high accuracy and rapid turnaround time support its use in resource-limited settings.

**Data availability statement:** De-identified, individual participant data from this study will be available to researchers whose proposed purpose of use is approved by the MORU data access committee. Enquiries or requests for data can be sent to datasharing@tropmedres.ac.

**Funding:** This research was conducted by the MORU Tropical Health Network, which is funded through a core grant block award from the Wellcome Trust (220211/Z/20/Z). For the purpose of open access, the author has applied a CC BY public copyright license to any Author Accepted Manuscript version arising from this submission. The funder had no role in study design, data collection and analysis, decision to publish, or preparation of the manuscript.

**Competing interests:** I have read the journal's policy and the authors of this manuscript have the following competing interests: DABD acts as a paid independent consultant for InBios. All other authors have declared that no competing interests exist.

## Author summary

Melioidosis is a serious infection caused by the soil-dwelling bacterium *Burkholderia pseudomallei*. Diagnosis requires culture of specimens taken from infected sites, which is the gold standard, but is often slow or unavailable in resource-limited settings. In our study, the Active Melioidosis Detect Plus rapid test was added into routine diagnostic processes. We found children received appropriate treatment faster, with the majority undergoing further testing to look for systemic infection. We also found that the rapid test had excellent accuracy when compared to culture.

## Introduction

Melioidosis is an important endemic infection prevalent in tropical and subtropical regions, caused by *Burkholderia pseudomallei*, a Gram-negative, soil-dwelling saprophyte [1]. The diagnosis of melioidosis requires microbiological confirmation due to clinical features that overlap with those of other bacterial infections. Traditional culture-based diagnostic methods, which remain the gold standard, typically take 48 hours or longer to yield results [2–4]. This delay presents critical challenges, particularly because *B. pseudomallei* exhibits intrinsic resistance to many commonly used first-line antibiotics and therefore requires specific treatment. Delaying the initiation of appropriate antimicrobial therapy can compromise patient outcomes.

In resource-limited settings, patients with skin and soft tissue infections without systemic symptoms are frequently treated by surgical debridement and discharged with empirical antibiotics targeting common pathogens such as *Staphylococcus aureus* and *Streptococcus* spp., which are ineffective against *B. pseudomallei*. Microbiological culture is not always available and, even when performed, results are often delayed, by which time patients are often uncontactable (common reasons include living far from hospitals with capacity for microbiological culture, peripatetic lifestyles, and unreliable access to a functioning telephone). Additionally, such patients are less likely to be thoroughly screened (with blood culture and abdominal imaging) for disseminated disease, which can lead to serious complications if left undetected. Current guidelines for melioidosis treatment recommend an initial intensive phase of treatment with intravenous antibiotics (ceftazidime or meropenem) for at least 10–14 days [5], yet in low-resource healthcare settings, cutaneous melioidosis is often managed with oral antibiotics (co-trimoxazole) alone [2]. This makes exclusion of disseminated disease imperative to prevent sub-optimal treatment.

The Active Melioidosis Detect Plus rapid diagnostic test (AMD-RDT) is a qualitative, membrane-based lateral flow immunoassay point-of-care test that detects the capsular polysaccharide of *B. pseudomallei,* developed by InBios (Seattle, WA, USA) [6]. Previous studies of the AMD-RDT have shown promise, demonstrating high sensitivity and near-perfect specificity for pus specimens when compared to culture [3,7]. This rapid test significantly reduces diagnostic turnaround time, offering

an advantage where timely diagnosis is critical, especially in resource-limited settings where culture may not be readily available. However, the impact of the test on clinical management decisions remains underexplored. This study assessed the impact of the AMD-RDT on the management of children presenting with skin and soft tissue infections to a paediatric hospital in northern Cambodia.

## Methods

### Ethics statement

Written informed consent was not sought because specimens were collected as part of routine care at the discretion of the surgical team. However, the study was explained to patients and/or their caregivers, including their right to opt out of the study if they wished to do so (see Intervention section). The study was approved by the AHC Institutional Review Board (IRB-0099/23AHC; March 2023), the Cambodian National Ethics Committee for Human Research (NECHR149; May 2023), and the Oxford Tropical Research Ethics Committee (OxTREC: 528:23; July 2023). The study is reported in line with the CONSORT guidelines for feasibility trials (S1 Text) [8].

### Study setting

The study was conducted at Angkor Hospital for Children (AHC), Siem Reap, Cambodia. AHC is a non-governmental paediatric healthcare organisation with a nationwide catchment area. The hospital provides services from primary to tertiary care and has 89 beds situated on two medical wards, a surgical ward, a special care baby unit, and neonatal and paediatric intensive care units. The 12-bed surgical unit has approximately 700 annual admissions and a minor procedure room which provides low-risk surgical care to approximately 10,000 patients with 1,600 procedures conducted annually. Most children presenting with skin and soft tissue infections are treated in the minor procedure room, receiving wound management and/or incision and drainage of abscesses, before being discharged home the same day.

### Study design

This was an interventional ambispective cohort study with a before-and-after comparison. The study included both a retrospective and a prospective arm; with recruitment for the prospective arm beginning after the intervention, i.e., introduction of the AMD-RDT. For the retrospective arm, clinical and laboratory data were analysed from children with pus or pus swab specimens sent to the microbiology laboratory between 14th July 2022 and 13th July 2023 that were culture-positive for *B. pseudomallei*.

For the prospective arm, consecutive pus and pus swab specimens sent to the microbiology laboratory with a request for melioidosis culture from 14th July 2023 until 31st December 2024 were screened by laboratory staff to identify potential participants. Participants were included if they were aged under 16 years and were presenting with a skin or soft tissue infection (including cellulitis, wound infections, skin abscesses, lymphadenitis, or parotitis). Pus or pus swab specimens from which the initial Gram stain showed Gram-positive cocci only were excluded (S1 Fig). Patients were not enrolled if they had already been included in the study for the same illness episode (still receiving antibiotics or under active follow up).

### Data collection

A structured paper case report form was used to capture data for both study arms. Data were subsequently entered into the study's REDCap database hosted at the Cambodia Oxford Medical Research Unit, Angkor Hospital for Children [9]. Clinical and laboratory data were retrieved from the Laboratory Information Management System (LIMS), Hospital Information System (HIS), clinical records, and the hospital and laboratory logbooks. Data included basic demographics, clinical presentation, management plan (including diagnostic tests, disposition, and treatment), microbiological workup,

and results of the AMD-RDT. All variables were prospectively defined in a data dictionary to ensure consistency of interpretation across the research team.

### Intervention

Prior to commencement of the prospective arm, a meeting with the surgical team was held to introduce the study and explain the eligibility criteria, AMD-RDT procedures, and the interpretation of negative and positive AMD-RDT results. During the meeting, the preliminary nature of the rapid test results was emphasised and that the final diagnosis should be based on the culture results. It was explained that the initial test results (Gram stain with or without AMD-RDT) would be available approximately one hour after the specimens were collected.

While waiting for their results, patients and/or caregivers had the study explained to them, were given time to ask questions, and had the opportunity to opt out. The study team phoned the surgical team to inform them of the initial Gram stain result and, if performed, the AMD-RDT result. All patient management decisions were the responsibility of the surgical team who had access to local hospital and national treatment guidelines for melioidosis.

### Laboratory procedures

The AMD-RDT was performed on pus or pus swab specimens for participants in the prospective arm, with results reported back to the treating surgical team approximately 60 minutes after specimen collection. Following pre-test steps for pus or pus swab specimens, the test was performed following the manufacturer's instructions which was described previously [3].

For pus specimens, a heating step, as described in a recent evaluation of the test (and intended for inclusion in the manufacturer's FDA approval application) [3], was performed prior to conducting the AMD-RDT to enhance sensitivity. Briefly, 30 μL of pus was collected using a pipettor and transferred into a 2 mL sterile microtube. Four drops of 20x lysis buffer were added, and the microtube was vortexed thoroughly in three 10-second rounds. The microtube was then placed in a heating block set to 95°C for 15 minutes, with brief vortexing at 5 and 10 minutes. For pus swab specimens, prior to performing the AMD-RDT, the swabs were expressed in a microtube containing 500 μL of sterile water. A 50 μL aliquot of the swab eluate was then transferred into a new 2 mL sterile microtube. Four drops of 20x lysis buffer were added, and the microtube was vortexed for three 10-second intervals. Following the pre-test steps, 50 μL of the prepared pus or pus swab specimens was applied to the AMD-RDT sample port, followed by the addition of two drops of chase buffer. The results were read and interpreted after 15–20 minutes at room temperature.

Following Gram stain (with or without AMD-RDT), routine and *B. pseudomallei* culture was performed on all pus and pus swab specimens using blood agar, chocolate agar, MacConkey agar, Ashdown agar, and in-house selective broth (S2 Text) that was subcultured onto Ashdown agar (S1 Fig). *B. pseudomallei* isolates were identified using latex agglutination (Mahidol University, Thailand) and by MALDI-TOF mass-spectrometry (bioMerieux VITEK MS) using in-house SuperSpectra [10,11]. The microbiology laboratory obtained ISO accreditation under standards ISO 15189 and ISO 15190 in November 2023.

### Outcomes

There were two co-primary outcomes. The first was time to initiation of an antimicrobial with activity against *B. pseudomallei* for participants with culture-confirmed *B. pseudomallei* infection. This was defined as the time between collection of the pus specimen and the start of an antimicrobial agent with activity against *B. pseudomallei* (ceftazidime, meropenem, or co-trimoxazole). The second co-primary outcome was the proportion of participants with culture-confirmed *B. pseudomallei* infection in whom a blood culture and abdominal ultrasound were completed. The secondary outcome was the diagnostic accuracy of the AMD-RDT on pus/pus swab specimens, using microbiological culture as the reference standard.

## Sample size

Sample size calculation was pragmatic and based on the resources available for the study which permitted prospective recruitment for 12 months, which was extended to 15 months during the study due to a slower than expected recruitment rate. Using routinely collected baseline data from previous years at the same hospital, it was expected that 100 participants would be recruited over 12 months, of which approximately 40 were anticipated to have *B. pseudomallei* identified. Retrospective cases spanned one calendar year prior to the introduction of the AMD-RDT.

## Statistical analysis

All variables were summarised and reported using descriptive statistics. Continuous variables were reported using mean with standard deviation or median with $25^{th}$ and $75^{th}$ centiles. Categorical variables were summarised with frequency and percentage. Comparisons were made using the Mann-Whitney-U or Fisher's Exact test.

Kaplan-Meier analysis and Cox regression were used to compare the time from pus collection to initiation of an antimicrobial with activity against *B. pseudomallei* between the prospective and retrospective arms. Participants who received antibiotics with activity against *B. pseudomallei* empirically (i.e., prior to pus collection) were excluded from the time to effective treatment analysis. Time 0 was set as the time when the pus specimen was collected. Children who did not receive effective treatment were censored at maximum follow up time. Hazard ratios with 95% confidence intervals are reported.

In the prospective arm, the accuracy of the AMD-RDT compared to culture was determined by the area under the receiver operating characteristic curve (AUC), sensitivity, specificity, positive predictive value (PPV), and negative predictive value (NPV). These metrics were calculated with 95% confidence intervals using Stata version 18 (Stata Corp LLC, College Station, TX, USA) and R version 4.3.2.

# Results

## Retrospective arm

Between $14^{th}$ July 2022 and $13^{th}$ July 2023, 476 pus and pus swab specimens were collected from 408 patients and sent to the microbiology laboratory. Of those, 171 specimens included a request for melioidosis culture. Twenty-five (14.6%; 25/171) specimens from 25 unique participants grew *B. pseudomallei* (Fig 1). Median age was 5.4 years (interquartile range [IQR] 2.5 to 8.2 years) and 68.0% (17/25) of the cohort were male (Table 1). The most common clinical presentation was acute lymphadenitis (52.0%; 13/25), followed by skin infections (36.0%; 9/25), and parotitis (12.0%; 3/25). Fifteen (60%;15/25) participants came from within Siem Reap province, while the other ten (40%; 10/25) came from four other provinces (S2 Fig). Sixty percent (15/25) of participants were confirmed as having recovered after receiving the recommended three-month course of antibiotics, while 32.0% (8/25) were lost to follow-up prior to completion of treatment; all were prescribed at least one month of oral co-trimoxazole after diagnosis. Two participants (2/25; 8.0%) did not receive antibiotics with efficacy against *B. pseudomallei*, one was treated with surgical drainage alone and the other discharged with antibiotics without activity against *B. pseudomallei* prior to culture results being available.

## Prospective arm

There were 208 pus and pus swab specimens sent with a request for melioidosis culture to the microbiology laboratory from 202 unique patients between $14^{th}$ July 2023 and $31^{st}$ December 2024. Gram stain of the specimens identified 100 specimens with Gram-positive cocci (GPC) only, which were excluded. One of two specimens that were sent from the same patient was excluded. The remaining 107 specimens from 107 unique patients were enrolled. No participants opted out of the study. Sixty-nine specimens (64.5%; 69/107) had no bacteria seen on Gram stain, Gram-negative bacilli (GNB)

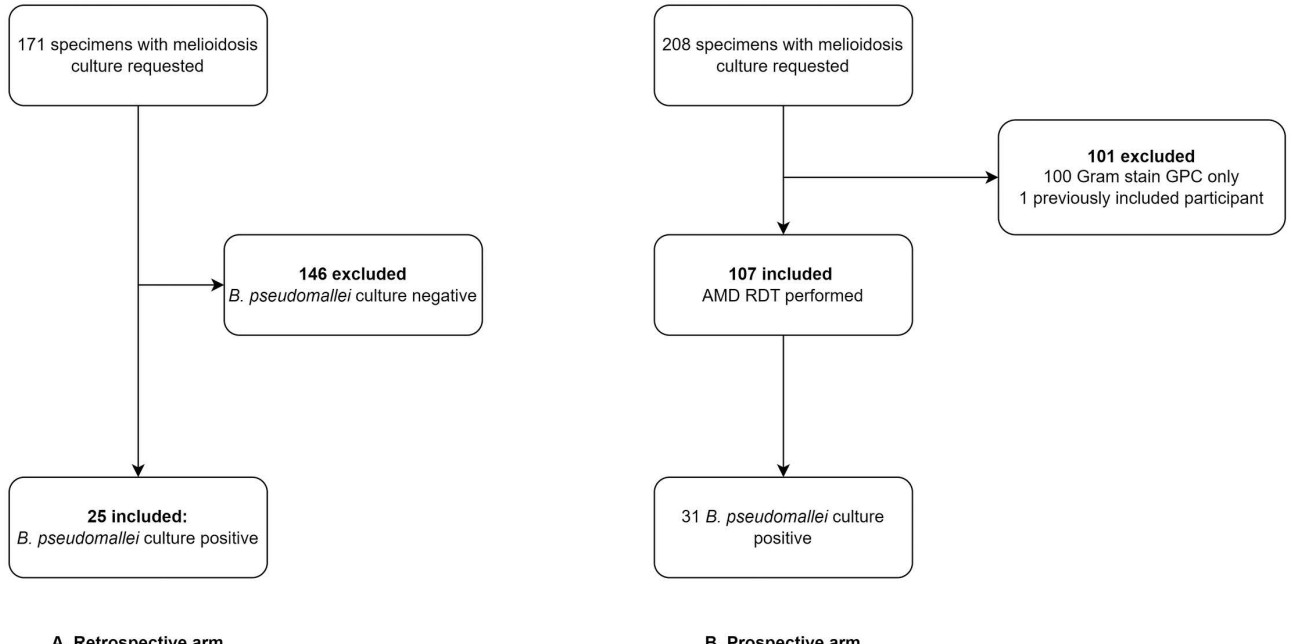

**Fig 1. Enrolment flowchart for retrospective (A) and prospective (B) arms.** Culture positive specimens were included in the time to effective treatment and diagnostic workup analyses. All prospective specimens (n = 107) were included in the AMD-RDT diagnostic accuracy analyses. GPC = Gram-positive cocci.

were seen in 23 specimens (21.5%; 23/107), and mixed GPC and GNB were seen in 15 specimens (14.0%; 15/107) (S1 Table).

Thirty-one of the 107 specimens (28.9%) included in the study were culture positive for *B. pseudomallei* (Fig 1). Median age of children who were positive for *B. pseudomallei* was 3.9 years (IQR 3.6 to 6.8 years) and 58.1% were female (18/31). Fifteen specimens (48.4%; 15/31) were sent from participants with parotitis, twelve (38.7%; 12/31) from participants with acute lymphadenitis, and four (12.9%; 4/31) from participants with skin infections (Table 1). Twenty-one participants (67.7%; 21/31) came from within Siem Reap province while the other ten (32.3%; 10/31) came from three other provinces (S2 Fig).

Twenty-two participants (71.0%; 22/31) were confirmed as having recovered after receiving the recommended three months of antibiotics. Nine participants (29.0%; 9/31) were lost to follow-up prior to completion of three months therapy; but all were prescribed at least one month of co-trimoxazole after diagnosis.

## Time to effective treatment

In the retrospective arm, all participants received empirical treatment for their infection. Seven (28%; 7/25) received empirical antibiotics with activity against *B. pseudomallei* prior to pus collection; 16 were switched to effective melioidosis treatment following specimen collection and two, both presenting with skin infections, did not receive antimicrobials with activity against *B. pseudomallei* (S3A Fig in S3 Fig). Of the participants who received treatment active against *B. pseudomallei*, seven (30.4.0%; 7/23) received intensive phase treatment (intravenous ceftazidime) followed by oral co-trimoxazole, while 16 (69.6%; 16/23) received oral co-trimoxazole monotherapy.

In the prospective arm, all *B. pseudomallei* culture-positive participants received empirical treatment for their infection episodes. Of those, eleven (35.5%; 11/31) received antimicrobials with activity against *B. pseudomallei* prior to

**Table 1. Characteristics of participants with confirmed melioidosis in the retrospective and prospective arms.**

| Characteristics | Retrospective arm N (%), Total = 25 | Prospective arm N (%), Total = 31 | p-value |
|---|---|---|---|
| **Median age** (IQR), years | 5.4 (2.5-8.2) | 3.9 (3.6-6.8) | 0.85 |
| **Sex** | | | |
| Female | 8 (32.0) | 18 (58.1) | 0.064 |
| Male | 17 (68.0) | 13 (41.9) | |
| **Clinical presentation** | | | |
| Parotitis | 3 (12.0) | 15 (48.4) | 0.009 |
| Skin infection | 9 (36.0) | 4 (12.9) | |
| Lymphadenitis | 13 (52.0) | 12 (38.7) | |
| **Gram stain of pus/pus swab** | | | |
| No bacteria seen | 24 (96.0) | 10 (32.3) | < 0.001 |
| Gram-negative bacilli only | 1 (4.0) | 19 (61.3) | |
| Mixed | 0 | 2 (6.5) | |
| **Diagnostic workups** | | | |
| Blood culture performed | 7 (28.0) | 25 (80.6) | < 0.001 |
| *B. pseudomallei* isolated | 0 | 2 | |
| Abdominal ultrasound performed | 6 (24.0) | 29 (93.5) | < 0.001 |
| Splenic abscess | 2 | 2 | |
| Liver abscess | 0 | 2 | |
| **Antimicrobial management** | | | |
| No antimicrobials with activity against *B. pseudomallei* | 2 (8.0) | 0 | |
| Empirical anti-*B. pseudomallei* treatment | 7 (28.0) | 11 (35.5) | 0.58 |
| Median time to effective antimicrobial treatment (IQR), hours* | 118.4 (21.3-143.3) | 14.4 (3.3-70.7) | 0.057 |
| **Outcome** | | | |
| Recovered** | 16 (64.0) | 22 (71.0) | 0.77 |
| Lost to follow-up*** | 9 (36.0) | 9 (29.0) | |

* Includes n = 18 from retrospective and n = 20 from prospective arm who did not receive effective treatment before pus collection. Time to treatment was set to maximum follow up time for the two children in the retrospective arm who did not receive antibiotics with efficacy against *B. pseudomallei*. ** One participant was treated with surgical debridement alone and considered recovered at follow-up. *** One participant was treated with antibiotics without activity against *B. pseudomallei* and was uncontactable after leaving the study site.

pus collection and 20 (64.5%; 20/31) were switched to effective treatment following pus collection (S3B Fig in S3 Fig). Twenty-one participants (67.7%; 21/31) received intensive phase treatment prior to oral therapy, whilst 10 (32.3%; 10/31) received oral co-trimoxazole monotherapy.

There was no difference in the frequency of effective empiric treatment before pus collection between the prospective and retrospective arms (11/31 (35.5%) v. 7/25 (28.0%), p = 0.58). Among the 18 children in the retrospective arm and 20 in the prospective arm who did not receive effective empiric treatment before pus collection, median time from pus collection to initiation of treatment with an antimicrobial with activity against *B. pseudomallei* was shorter in the prospective arm but this was not statistically significant (118.4 hours [IQR 21.3 to 143.3 hours] in the retrospective arm and 14.4 hours [IQR 3.3 to 70.7 hours; p = 0.057] in the prospective arm; Hazard ratio 0.53; 95% CI 0.27 to 1.03; p = 0.061; Fig 2).

Among participants who received anti-*B. pseudomallei* antibiotics after pus collection, 10 (50.0%; 10/20) in the prospective arm and four (25.0%; 4/16, p = 0.17) in the retrospective arm received them on the day of pus collection, while four (20.0%; 4/20) in the prospective arm and eight (50%; 8/16, p = 0.081) in the retrospective arm received them more than three days after pus collection.

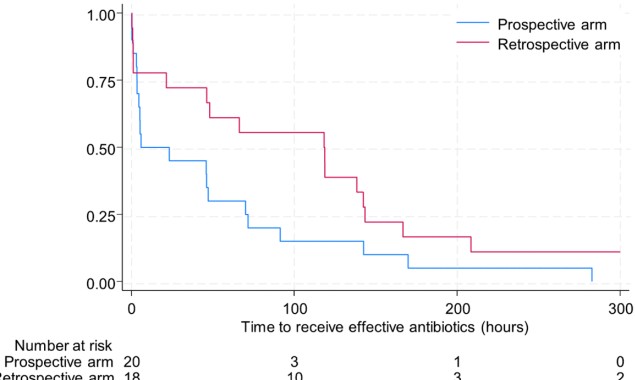

**Fig 2. Kaplan-Meier plot of the time to effective treatment for both study arms.** Included retrospective arm (n = 18) and prospective arm (n = 20). Time to effective treatment was defined from time of pus collection to initiation of antibiotics with activity against *B. pseudomallei*.

## Diagnostic workups

Both blood culture and abdominal ultrasound were performed in 24.0% (6/25) of participants in the retrospective arm and 80.6% (25/31) in the prospective arm (p < 0.001).

In the retrospective arm, blood cultures were performed in seven out of 25 participants (28.0%), all of which were negative, while abdominal ultrasound was conducted in six participants (24.0%; 6/25), identifying splenic microabscesses in two cases (Table 1). In the prospective arm, blood cultures were collected in 25 out of 31 participants (80.6%; p < 0.001), with *B. pseudomallei* isolated in two cases. Abdominal ultrasound was performed in 29 participants (93.5%; 29/31; p < 0.001), revealing intra-abdominal abscesses in three participants: one with a splenic abscess, one with a liver abscess, and one with both liver and splenic abscesses.

## Accuracy of the AMD-RDT

The AMD-RDT was performed on 107 specimens from the prospective arm. Among 31 culture positive specimens (four pus specimens and 27 pus swab specimens), 28 had a positive AMD-RDT result. All three negative AMD-RDT results occurred in pus swabs. None of the culture negative specimens had positive AMD-RDT results (Table 2).

Receiver operating characteristic curve analysis showed an AUC of 0.95 (95% CI 0.89-1.00), with a sensitivity of 90.3% (95% CI 74.2 to 98.0%), a specificity of 100% (95% CI 95.3 to 100%), a PPV of 100% (95% CI 87.7 to 100%), and a NPV of 96.3% (95% CI 89.4 to 99.2%).

The three pus swab specimens that were culture positive for *B. pseudomallei* but tested negative on the AMD-RDT were sent from two participants with acute parotitis and one participant with subacute cervical lymphadenitis. Information regarding antibiotic use prior to presentation was not available. Gram stain of the specimens did not identify presence of bacteria in two while Gram-negative bacilli were seen on one. Two specimens yielded a single colony of *B. pseudomallei*

**Table 2. Results of culture and AMD-RDTs for specimens enrolled in the prospective arm.**

| | | AMD RDT | | Total |
|---|---|---|---|---|
| | | **Positive** | **Negative** | |
| ***B. pseudomallei* culture** | **Positive** | 28 | 3 | **31** |
| | **Negative** | 0 | 76 | **76** |
| **Total** | | **28** | **79** | **107** |

on chocolate agar at 24 hours, with growth observed on all other culture media at 48 hours of incubation. The third swab grew *B. pseudomallei* on Ashdown agar after subculture from selective broth at 48 hours.

## Discussion

Melioidosis remains a significant diagnostic challenge in many endemic tropical regions due to limitations in laboratory capacity and prolonged turnaround time of culture-based diagnosis. To our knowledge, this is the first study assessing the impact of the AMD-RDT on the care of children presenting with suspected skin and soft tissue melioidosis. Using a pragmatic before-and-after study design, we found a reduction in time from pus collection to initiation of effective antibiotics after the AMD-RDT was incorporated into the routine diagnostic workflow. The proportion of children receiving effective antibiotics on the day of presentation increased by more than 50% (21/31; 67.7% vs. 11/25; 44.0%), with the proportion of children who never received an effective antibiotic dropping from close to one in ten to zero after the rapid test was introduced.

Nevertheless, even though the AMD-RDT took less than one hour to perform, only two-thirds of the participants (21/31) received effective antibiotics on the day of sample collection, and 10% (4/31) received effective antibiotics more than three days later. Delays were often observed in participants whose specimens were collected towards the end of a day, with results only being reviewed the next working day, which could be up to three days for specimens collected prior to a weekend. This illustrates the importance of considering the existing patient care pathway when introducing a new diagnostic test, including ensuring that there is a mechanism for results to be acted upon, otherwise the potential for impact will be limited. This observation is not unique to melioidosis. Similar findings have been reported in a meta-analysis of studies investigating interventions aimed at improving the time to effective antibiotics in bacteraemic patients [12].

After the training session and the introduction of the AMD-RDT, we saw a three-fold increase in the proportion of participants who underwent workup for disseminated disease. Two participants with bacteraemia and three with intra-abdominal abscesses were identified in the prospective arm, whose disseminated disease may have gone unrecognised leading potentially to unfavourable outcomes. In addition, the proportion of participants who received the recommended two-phase therapy doubled. These findings highlight the potential benefit of using the AMD-RDT, which provides same day diagnosis, together with appropriate training to enable timely and targeted treatment of melioidosis, including for patients with systemic involvement.

We found that the test had excellent sensitivity and specificity and demonstrated high diagnostic accuracy as shown by the AUC. These findings are consistent with those reported by Currie and colleagues who refined the sample preparation steps (heating for pus and lysis buffer for swabs to enhance test sensitivity) that we subsequently used in our study [3]. Our results are also aligned with previous studies in term of test specificity [4,7,13,14]. Taken together, the evidence to date supports the use of the AMD-RDT on pus specimens collected from both adults and children. The three false negative AMD-RDT results were observed in specimens where bacterial load was likely low, as evidenced by their specimen culture result. We also observed that despite recommended by the microbiology laboratory to collect pus samples, pus swabs are often sent by the treating clinical teams.

This study has limitations. First, the results may have been confounded by temporal changes in healthcare worker behaviour. Nevertheless, a before-and-after design is an appropriate approach for evaluating implementation of an intervention with proven accuracy into routine care [15]. Second, it is unclear how much the increase in systemic diagnostic work ups was influenced by the surgical team training provided during implementation, rather than early diagnosis of melioidosis by the AMD-RDT prompting clinicians to investigate for systemic disease. Although empirical prescribing practices did not appear to alter, it is important to ensure that the rollout of any new test is supported by appropriate diagnostic stewardship. Third, although the pre-test steps may have improved the accuracy of the test, they limit its use to a laboratory setting with appropriate safety measures. Fourth, a higher than anticipated empirical prescribing rate of anti-*B. pseudomallei* antibiotics resulted in the group available for the time to effective treatment analysis being smaller than expected, and

whilst it did indicate a reduction in time to effective therapy (118 vs 14 hours), the estimate was imprecise and did not reach statistical significance. There were also a large number of patients lost to follow-up (29% in the prospective arm and 36% in the retrospective arm) which limits our ability to determine final clinical outcome for both groups, although this did not impact the study's co-primary outcomes. Finally, differences in the baseline characteristics, in particular a higher proportion of samples with bacteria visible on Gram stain in the prospective cohort which may reflect higher bacterial loads, may have introduced bias, which is a limitation of this non-randomised study design. Another study design limitation was that the study was conducted in a single-centre and in a paediatric only population, which may limit generalisability of the findings.

In conclusion, introduction of the AMD-RDT was associated with a reduction in time to effective antibiotic therapy and, together with training of the clinical staff, an increase in the proportion of participants completing a comprehensive diagnostic workup for systemic involvement. With excellent accuracy, the test is applicable for both children and adults in resource-limited settings where access to culture diagnosis may not be feasible or delayed. Where possible, the test should be an adjunct to, rather than a replacement for, culture-based diagnosis, as this could lead to loss of critical information on antimicrobial susceptibility over time. To maximise impact, it is imperative that the test is embedded within functioning care pathways that allow timely action following the result. Future research should look to replicate our findings and explore the cost-effectiveness of incorporating the AMD-RDT into routine diagnostic protocols in resource-limited settings.

## Supporting information

**S1 Fig. Diagnostic workflow for participants in the prospective arm.** Different media types and incubation conditions were used to allow detection of a broad range of pathogens with varying culture requirements. GPC: Gram-positive cocci; AMD-RDT: Active Melioidosis Detect Plus rapid diagnostic test; *B. pseudomallei*: *Burkholderia pseudomallei;* MALDI-TOF MS: Matrix-assisted laser desorption/ionisation mass spectrometry.
(TIF)

**S2 Fig. Case Distribution in Cambodia by Provinces and District of their origin.** (source of the basemap shapefile: https://data.humdata.org/dataset/cod-ab-khm).
(TIF)

**S3 Fig. Number of participants receiving empirical antibiotics with activity against *Burkholderia pseudomallei*.** A: Retrospective arm, number of participants = 25; B: Prospective arm, number of participants = 31.
(TIF)

**S1 Text.** Summary of information reported in accordance with CONSORT 2010 checklist
(source of the checklist: https://www.equator-network.org/reporting-guidelines/consort-2010-statement-extension-to-randomised-pilot-and-feasibility-trials/).
(DOCX)

**S2 Text. In-house media preparation: A.** Selective Broth, B: Ashdown Agar.
(DOCX)

**S1 Table. Characteristics of participants in prospective arm.**
(DOCX)

## Acknowledgments

We are grateful to all the participants and their caregivers, the AHC surgical team, and the COMRU laboratory and research staff for their specimen processing, data management and study monitoring. We acknowledge the support of InBios, who provided the AMD-RDTs used in the study. After preparation of the initial draft, the corresponding author (KS)

used the free version of OpenAI's ChatGPT (GPT-4o, May 2024 version) in order to enhance readability and grammar. After using this tool, all authors reviewed and edited the content as needed and take full responsibility for the content of the publication.

## Author contributions

**Conceptualization:** Keang Suy, Paul Turner, Clare L Ling, Arjun Chandna.

**Data curation:** Keang Suy, Seda Bott, Sophanith Real.

**Formal analysis:** Keang Suy, Sue J Lee.

**Investigation:** Keang Suy, Sona Soeng, Clare L Ling.

**Methodology:** Keang Suy, Sue J Lee, Paul Turner, Clare L Ling, Arjun Chandna.

**Project administration:** Keang Suy, Clare L Ling, Arjun Chandna.

**Resources:** Keang Suy, Nara Leng, Vuthy Sar, Sona Soeng, David AB Dance, Sue J Lee, Paul Turner, Clare L Ling, Arjun Chandna.

**Supervision:** Paul Turner, Clare L Ling, Arjun Chandna.

**Visualization:** Keang Suy, Sue J Lee, Clare L Ling.

**Writing – original draft:** Keang Suy.

**Writing – review & editing:** Keang Suy, Seda Bott, Nara Leng, Vuthy Sar, Sona Soeng, Sophanith Real, David AB Dance, Sue J Lee, Paul Turner, Clare L Ling, Arjun Chandna.

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
