## [Decision Letter · Decision Letter 0]

28 Oct 2025

Rapid diagnosis of skin and soft tissue melioidosis in children

Dear Dr. Suy,

Thank you for submitting your manuscript to PLOS Neglected Tropical Diseases. After careful consideration, we feel that it has merit but does not fully meet PLOS Neglected Tropical Diseases's publication criteria as it currently stands. Therefore, we invite you to submit a revised version of the manuscript that addresses the points raised during the review process.

Please submit your revised manuscript within 60 days. If you will need more time than this to complete your revisions, please reply to this message or contact the journal office at plosntds@plos.org. Please include the following items when submitting your revised manuscript:

We look forward to receiving your revised manuscript.

Kind regards,

Husain Poonawala

Academic Editor

Ana LTO Nascimento

Section Editor

Shaden Kamhawi

co-Editor-in-Chief

Paul Brindley

co-Editor-in-Chief

**Additional Editor Comments :**

The outcomes and analysis plan do not seem to align. Is the definition of "antimicrobials with activity against B. pseudomallei" not similar to "effective treatment following pus collection"? This is a distinction without a difference.

I would like to see more explanation of why time to effective treatment was chosen. I don't understand the rationale for separating out analysis by whether they got empiric treatment or not.

The sample sizes for the K-M curves are quite small - is there a reason that the analysis separates those who received empiric antibiotics from those who did not? Can you perform an analysis that uses the time from any antibiotic administration, perhaps only excluding those who received antibiotics prior to presentation at the hospitial? The effect sizes are quite wide

If a statistician is not already involved, I would recommended obtaining their help.

I am not sure how the blood culture and ultrasound workups are relevant to performance of the LDT - it seems out of place from the other two aims. It would be more relevant towards the role of diagnostic pathways in the workup of B. pseudomallei.

Lines 263 and 224 - I think it would be more accurate to say that all were prescribed a month of antibiotics. Received implies that they took the medication, but that cannot be confirmed since they were lost to follow-up.

The pus swabs were probably negative because swab is not a good sample collection device and there was likely very little specimen, highlighting the need for good sample collection as part of the diagnostic workup (and the larger issue that swabs should not be used at all)

I consider the 33% loss to follow-up a significant limitation and recommend including it in the discussion.

Is the real finding of this study that patients with B. pseudomellei require more thorough workup on initial presentation? I think the diagnostic performance of the test is a minor part of the improved outcomes, and what really made a difference was the training of the surgical team and improved diagnostic workup. My suggestion is to consider reframing the article on the role of all the changes that helped improve the diagnosis.

I would like to respectfully point out to the authors that submission of any material into ChatGPT results in a loss of confidentiality and to use AI tools for research with a full understanding of what happens to the information once submitted there.

**Journal Requirements:**

At this stage, the following Authors/Authors require contributions: Keang Suy, Seda Bott, Nara Leng, Vuthy Sar, Sona Soeng, Sophanith Real, David AB Dance, Sue J Lee, Paul Turner, Clare L Ling, and Arjun Chandna. Please ensure that the full contributions of each author are acknowledged in the "Add/Edit/Remove Authors" section of our submission form.

- TM on pages: 2, 3, 4, and 25.

Potential Copyright Issues:

i) Please confirm (a) that you are the photographer of the photos of Diagram of MPD Plus Cassette and Diagram of Negative, and Posotive and Invalid Test results in Supplementary S3 Text, or (b) provide written permission from the photographer to publish the photo(s) under our CC BY 4.0 license.

ii) Figure S4. Please (a) provide a direct link to the base layer of the map (i.e., the country or region border shape) and ensure this is also included in the figure legend; and (b) provide a link to the terms of use / license information for the base layer image or shapefile. We cannot publish proprietary or copyrighted maps (e.g. Google Maps, Mapquest) and the terms of use for your map base layer must be compatible with our CC BY 4.0 license.

5) For studies involving third-party data, we encourage authors to share any data specific to their analyses that they can legally distribute. PLOS recognizes, however, that authors may be using third-party data they do not have the rights to share. When third-party data cannot be publicly shared, authors must provide all information necessary for interested researchers to apply to gain access to the data. For more information, see:

https://journals.plos.org/plosntds/s/data-availability#loc-acceptable-data-access-restrictions

3) If any authors received a salary from any of your funders, please state which authors and which funders.

7) Your current Financial Disclosure states, "This research was funded in whole, by the Wellcome Trust [220211/Z/20/Z]. However, your funding information on the submission form doesn't indicate any funds. Please amend the Funding Information tab ensuring that the funders and grant numbers match between the Financial Disclosure field and the Funding Information tab in your submission form. Note that the funders must be provided in the same order in both places as well.

8) Please amend your 'Competing Interests' statement, and declare all competing interests beginning with the statement "I have read the journal's policy and the authors of this manuscript have the following competing interests:"

Note: If there are no competing interests to declare, please state "The authors have declared that no competing interests exist".

**Reviewers' Comments:**

Reviewer's Responses to Questions

**Key Review Criteria Required for Acceptance?**

**Methods**

-Are the objectives of the study clearly articulated with a clear testable hypothesis stated?

-Is the study design appropriate to address the stated objectives?

-Is the population clearly described and appropriate for the hypothesis being tested?

-Is the sample size sufficient to ensure adequate power to address the hypothesis being tested?

-Were correct statistical analysis used to support conclusions?

-Are there concerns about ethical or regulatory requirements being met?

Reviewer #1: 1.The manuscript mentions that “routine and B. pseudomallei culture was performed using blood agar, chocolate agar, MacConkey agar, Ashdown agar, and selective broth that was subcultured onto Ashdown agar.” To improve reproducibility and clarity, please specify the composition or source of the selective broth used in this study (e.g., enrichment broth type, manufacturer, or reference). Was it an in-house selective enrichment medium or a commercial formulation?

2.In Supplementary Figure S1 (“Diagnostic workflow for participants in the prospective arm”), could the authors clarify the rationale for using a CO₂-enriched atmosphere for blood and chocolate agar, but aerobic (O₂) incubation for MacConkey agar?

Typically, blood and chocolate agars are incubated in 5% CO₂ to support the growth of fastidious organisms (e.g., Streptococcus, Haemophilus), whereas MacConkey agar is incubated aerobically for enteric Gram-negative bacteria. It would be helpful if the authors could confirm that this was the intended rationale or provide a brief justification in the Methods or figure legend for clarity and reproducibility.

Reviewer #2: The methods are clearly articulated and are appropriate for the research questions. A power calculation was not performed. While the reason for this is stated clearly, it is likely that the small size of study has contributed to a non-statistically significant result for the first co-primary outcome.

**Results**

-Does the analysis presented match the analysis plan?

-Are the results clearly and completely presented?

-Are the figures (Tables, Images) of sufficient quality for clarity?

Reviewer #1: The results are well presented and clearly demonstrate the clinical utility of the AMD-RDT in improving diagnostic timeliness and treatment initiation for suspected melioidosis.The findings convincingly support the test’s value as an adjunct to culture-based diagnosis in resource-limited settings.

Reviewer #2: The analysis is presented clearly and is easy to follow. For the first co-primary outcome, the difference in time to effective treatment was not statistically significant, and this could be stated more clearly in the results.

**Conclusions**

-Are the conclusions supported by the data presented?

-Are the limitations of analysis clearly described?

-Do the authors discuss how these data can be helpful to advance our understanding of the topic under study?

-Is public health relevance addressed?

Reviewer #1: The discussion section is well-written, coherent, and effectively links the study findings with existing literature. The authors have clearly articulated the significance of introducing the AMD-RDT in clinical workflows and provided a balanced interpretation of their results. The linkage between diagnostic turnaround time, clinical outcomes, and system-level implementation challenges is thoughtfully addressed.

Reviewer #2: I would suggest some additional discussion of aspects of the data that warrant consideration, as well as expanded discussion on potential limitations of the study.

There are significant differences in the baseline characteristics of the two groups (retrospective and prospective). These differences are presented in Table 1, but are not discussed at all. Do the authors have any theories for why: (1) more females were affected in the prospective cohort, and more males in the retrospective cohort; (2) parotitis made up nearly half of the cases in the prospective cohort, but only 12% of the cases in the retrospective cohort; (3) skin infections made up more than a third of the cases in the retrospective cohort, but only 12.9% of the prospective cohort; (4) GNBs were more identified on microscopy in 61.3% cases in the prospective cohort but only 4% of cases in the retrospective cohort? Is it possible that a higher rate of parotitis and lower rate of skin infection may have contributed to seeing more positive microscopy, and that this might have inflated the sensitivity of the lateral flow assay?

Do the authors think it is possible that the sensitivity of the lateral flow assay (LFA) may be higher in samples that have a high enough bacterial load to be positive on Gram stain microscopy? If the LFA had been tested on the earlier cohort, where the vast majority of cases had no bacteria seen on microscopy, is it possible that the LFA may have performed less well, in terms of sensitivity? The authors should discuss this possibility, and acknowledge that the unbalanced cohorts introduces a risk of bias and that this is an important limitation.

The authors acknowledge the potential for the training to have contributed to higher rates of investigation for cases in the prospective arm (the second co-primary outcome). It would be good to see a little more discussion of this, as I suspect that training is more likely to lead to this kind of change in behaviour than a positive diagnostic test, especially in a cohort of patients who will already have a positive diagnostic test (albeit the culture result will come later). The training is probably also largely responsible for the higher proportion of patients getting intensive phase intravenous treatment. It would be interesting to see more discussion about whether or not this is always required.

While it does not appear to have been part of the analysis plan, a comparison of length of stay would be interesting to see. It may be that while the group who had the LFA test received a diagnosis and effective treatment sooner, they may also have had longer hospital stays, and the question of if the additional intravenous therapy is necessary or beneficial may not be clear.

**Editorial and Data Presentation Modifications?**

Reviewer #1: (No Response)

Reviewer #2: (No Response)

**Summary and General Comments**

Reviewer #1: (No Response)

Reviewer #2: This study is well designed and the results and discussion are well presented. The conclusions are sound, and appropriate based on the results of the analysis. However, further discussion about the limitations of the study design and the potential impacts of disparate before- and after- cohorts and training linked to introduction of the new test would be valuable.

PLOS authors have the option to publish the peer review history of their article (what does this mean? ). If published, this will include your full peer review and any attached files.

**Do you want your identity to be public for this peer review?** For information about this choice, including consent withdrawal, please see our Privacy Policy .

Reviewer #1: No

Reviewer #2: **Yes:** Joshua R Francis

**Figure resubmission:**
---

## [Decision Letter · Decision Letter 1]

20 Jan 2026

Dear Dr. Suy,

We are pleased to inform you that your manuscript 'Rapid diagnosis of skin and soft tissue melioidosis in children' has been provisionally accepted for publication in PLOS Neglected Tropical Diseases.

Best regards,

Husain Poonawala

Academic Editor

Ana LTO Nascimento

Section Editor

Shaden Kamhawi

co-Editor-in-Chief

Paul Brindley

co-Editor-in-Chief

Reviewer's Responses to Questions

**Key Review Criteria Required for Acceptance?**

**Methods**

-Are the objectives of the study clearly articulated with a clear testable hypothesis stated?

-Is the study design appropriate to address the stated objectives?

-Is the population clearly described and appropriate for the hypothesis being tested?

-Is the sample size sufficient to ensure adequate power to address the hypothesis being tested?

-Were correct statistical analysis used to support conclusions?

-Are there concerns about ethical or regulatory requirements being met?

Reviewer #1: No further revisions are required.

Reviewer #2: The methods are clearly articulated.

**Results**

-Does the analysis presented match the analysis plan?

-Are the results clearly and completely presented?

-Are the figures (Tables, Images) of sufficient quality for clarity?

Reviewer #1: No further revisions are required.

Reviewer #2: The changes to the results are good, and have strengthened the article.

**Conclusions**

-Are the conclusions supported by the data presented?

-Are the limitations of analysis clearly described?

-Do the authors discuss how these data can be helpful to advance our understanding of the topic under study?

-Is public health relevance addressed?

Reviewer #1: -No further revisions are required.

Reviewer #2: The discussion and conclusions are appropriate.

**Editorial and Data Presentation Modifications?**

Reviewer #1: No further revisions are required.

Reviewer #2: I have no suggestions for further modifications.

**Summary and General Comments**

Reviewer #1: No further revisions are required.

Reviewer #2: Congratulations to the authors on a worthwhile study and high quality manuscript.

PLOS authors have the option to publish the peer review history of their article (what does this mean? ). If published, this will include your full peer review and any attached files.

**Do you want your identity to be public for this peer review?** For information about this choice, including consent withdrawal, please see our Privacy Policy .

Reviewer #1: No

Reviewer #2: **Yes:** Prof Joshua Francis

---

## [Editor Report · Acceptance letter]

Dear Doctor Suy,

We are delighted to inform you that your manuscript, "Rapid diagnosis of skin and soft tissue melioidosis in children," has been formally accepted for publication in PLOS Neglected Tropical Diseases.

Best regards,

Shaden Kamhawi

co-Editor-in-Chief

Paul Brindley

co-Editor-in-Chief
